# Pesticide and Yeast Interaction in Alcoholic Fermentation: A Mini-Review

Kevin Becerra [1,2], Soumya Ghosh [3] and Liliana Godoy [1,*]

1 Departamento de Fruticultura y Enología, Facultad de Agronomía e Ingeniería Forestal, Pontificia Universidad Católica de Chile, Santiago 8940000, Chile
2 Centro de Investigación SIDAL LTDA, Casablanca 2480505, Chile
3 Department of Genetics, Faculty of Natural and Agricultural Sciences, University of the Free State, Bloemfontein 9301, South Africa
* Correspondence: liliana.godoy@uc.cl; Tel.: +56-22354-5729

**Abstract:** The current investigation briefly reviews previous studies about the fate of pesticides used in wine grape production during the alcoholic fermentation process, and how these could affect the correct functioning of yeast. The present review discusses the fact that yeasts could be used as a biological tool for pesticide dissipation, diminishing the concentration present in the grapes during the production process. The previous have never been directly boarded by other authors. The first part explores the influences of pesticides on yeasts and elucidates their effect on the fermentation process; also, some examples are analyzed of molecular studies involving the effect of pesticides on yeast. The second part discusses the effect of yeast on pesticide residues and their capacity to reduce its concentration during the alcoholic fermentation process, which varies among the different pesticides. In addition, this review discusses the mechanism by which yeast cells adsorb and/or degrade pesticides. In the last part, some examples of using yeasts as a possible remediation tool in wine and how the industry could use this to ensure consumers that a product is without pesticide residues are also discussed. This review shows that there is a natural capacity for the reduction of pesticide residue concentration by yeasts, and the effects of pesticides on yeast development is a variable phenomenon. This information guides advancement in pesticide removal from wine.

**Keywords:** pesticides; residues; wine; yeast; bioremediation

## 1. Introduction

Modern agriculture is designed to maximize profit by increasing the yields and quality of the products. One of the approaches to achieve this could be by reducing production costs and controlling the pest damage of the food crops, resulting in economical products for the consumers. In this regard, pesticides for agricultural use are mainly classified into three most important groups; fungicides (F), insecticides (I), and herbicides (H) play a pivotal role in controlling pests and diseases, either to control immediate infestations or to anticipate future problems [1]. Because of their extensive applications, a large amount of the applied pesticides find their way as 'residue' in the environment, especially into the terrestrial and aquatic food chains, where they could stem into long-term adverse health effects [2]. Each year, 140,000 tons of pesticides are sprayed into European Union (EU) crops alone. Although the pesticides start dissipating after spraying, the dissipation rate varies with the pesticide's physicochemical properties, type, food material, and environmental factors [3]. The pesticides used in agriculture require a time interval before harvesting in order to decrease their concentration to a level safe for human consumption; this last is known as maximum residue level (MRL). The time between application and harvest diverges between the sprayed pesticides and the type of crops. If the conditions of good agricultural practice (GAP) such as pesticide application and pre-harvest interval are not

met, harvested crops may contain residues above their MRL, and even when the pre-harvest interval is accomplished, residues may still be found.

The most important regulations around the world are the European Union regulations and the United States regulations, that act as a guide for the rest of the world. European Union pesticide regulation involves the European Food Safety Agency and the European Chemical Agency. The latter does not imply that countries that are members of the EU made their own MRLs. On the other hand, the United States' regulations are imposed by the Environmental Protection Agency. Regulations concerning pesticide residues in food worldwide consider several parameters, such as toxicity for humans and other nontarget species, environmental risk, and even the dietary information of the country. For the previous reasons, it is difficult to set or discuss specific tolerance, since a product such as wine is commercialized all over the world, having to go along with a wide range of tolerances. Table 1 shows the maximum residue limit in wine grapes for different pesticides.

**Table 1.** Maximum residue limit in wine grapes for different pesticides in mg/kg.

| Pesticide | EU [4] | US [5] | Chile [6] | Codex [7] |
|---|---|---|---|---|
| **Acetamiprid** | **0.5** | **0.35** | **0.5** | 0.5 |
| Azoxystrobin | 3 | 2 | 2 | 2 |
| Boscalid | 5 | 5 | 5 | 5 |
| Buprofezin | 0.01 | 1 | 1 | 1 |
| Chlorothalonil | 0.01 | * | 3 | 3 |
| Chlorpyrifos | 0.01 | * | 0.5 | 0.5 |
| Cyprodinil | 3 | 3 | 3 | 3 |
| Dimethomorph | 3 | 3 | * | 3 |
| Fenarimol | 0.3 | * | 0.3 | * |
| Fludioxonil | 4 | 2 | 2 | 2 |
| Folpet | 20 | 50 | 10 | 10 |
| Imidacloprid | 0.7 | 1 | 1 | 1 |
| Indoxacarb | 2 | 2 | 2 | 2 |
| Iprovalicarb | 2 | 2 | * | * |
| Lamda-cyhalotrin | 0.2 | 0.01 | 0.2 | * |
| Mepanipyrim | 2 | 1.5 | * | * |
| Metalaxyl | 1 | 2 | 1 | 1 |
| Penconazole | 0.5 | * | 0.4 | 0.4 |
| Procymidone | 0.01 | 5 | * | * |
| Pyrimethanil | 5 | 5 | 4 | * |
| Tebuconazole | 1 | 6 | 6 | 6 |
| Vinclozolin | 0.01 | 6 | * | * |

* MRL not established for wine grapes.

There are differences in the MRL between different regulations. For instance, Acetamiprid has a tolerance of 0.35 for the US and 0.5 for the rest of the regulations. Tebuconazole has six-fold less tolerance for the EU than for other regulations. Folpet shows tolerances of 50, 20, and 10 mg/kg for the US, EU, and Codex/Chile, respectively. In addition, in some regulations, some molecules have not established their MRL for wine grapes, such as Vinclozolin and Fenarimol in Codex (Table 1).

The presence of pesticide residues leads to rejection by the consumers, even with concentrations below the MRL. Consumers demand more information about this aspect; thus, it has become a regular feature for the agro-industry, specifically the wine industry. Many factors could affect wine market price, and pesticides are of major importance, negatively affecting batches and causing their entire losses. Despite the existence of legal regulations regarding pesticide residues allowing different concentration levels, the main goal for the wine industry is to have zero residues in wine. Consequently, the wine industry must develop new tools to produce wines without any pesticide residues.

Removing pesticides from wine has been approached using physical, chemical, and oenological methods [8]. The physical methods are pulsed electric field, ultrasounds, and

microfiltration, and have proved to strongly respond to pesticide properties, molecular structure, and membrane type [9]. Chemical methods are mainly related to the chemical adsorption removal of pesticides. Several chemical methods have been reported as ozone treatments [10] and others as related to compounds used in the filtration and clarification process in winemaking, such as activated carbon and bentonite [11]. Finally, among oenological methods are steps such as pressing, destemming, and alcoholic and malolactic fermentation. All the previous processes reduce, to a greater or lesser degree, pesticide residues. One common problem associated with oenological methods is the extraction of some important constituents of wine, such as aroma and color. Bentonite could reduce aroma compounds due to a nonselective capacity of remotion [8]. The use of activated carbon has demonstrated to be efficient in the reduction of ochratoxin A, but also had an impact on the color compounds of the wine, reducing flavonoid phenols and anthocyanins [12].

*Saccharomyces cerevisiae* is the principal yeast used in today's food and alcoholic beverage industries to produce bread, beer, spirits, cider, and wine. It is well-recognized that not all the *S. cerevisiae* strains are suitable for the fermentation process, and the ability to produce quality foods and beverages differs significantly among them [13]. Additionally, *S. cerevisiae* is the most important yeast in the wine industry. During alcoholic fermentation, yeast ferments sugars to ethanol and other compounds such as glycerol, acetate, succinate, and esters, which affect the sensorial properties of wine. Similarly, cell constituents such as polysaccharides or proteins are released from yeast cells after the autolysis process and become part of the wine composition [14].

Inoculating active dry yeasts during the winemaking process has become common in most wine-producing regions [15]. The growth of each wine yeast species is characterized by a specific metabolic activity, which determines concentrations of flavor compounds in the final product. Active dry yeast such as *S. cerevisiae* has been selected and differentiated from other yeast according to different characteristics such as high and low temperature tolerance, production and tolerance to $SO_2$, ethanol tolerance, pH tolerance, nitrogen requirements, and $H_2S$ production. However, it must be underlined that, within each species, significant strain variability has been recorded [16]. It has been reported that the type of yeast strain selected is a factor strongly affecting alcoholic fermentation, and, therefore, the quality of the wine [14].

Concerning pesticides, during the winemaking process, which includes destemming, pressing, and fermentation, there is a natural reduction of most of the pesticide residues [17]. However, it has been noted that alcoholic fermentation is one of the most important steps for pesticide dissipation [17–25], where the yeast species can degrade and adsorb the residues at various extents without altering their fermentative activities [26]. The removal process would be through the adsorption or degradation of the pesticides, being adsorption capacity as the main one [21].

The extent of the effects produced by pesticide residues in the wine industry can not only reach a legal problem, but also can in some cases directly affect the alcoholic fermentation process by reducing the yeast activity, affecting different fermentation parameters, and the production of some byproducts of the fermentation [27]. The explanation of how pesticides affect yeast has mostly been an indirect phenomenon of study, but there are a few investigations that have attempted to analyze this more profoundly.

## 2. Influence of Pesticide on Yeasts

Fermentation is a simple process where sugars are fermented mainly to alcohol and carbon dioxide [14]. Therefore, the potential of fermentation for improving the nutritional quality and safety of foods should be viewed within the context of the complete food processing operation. Additionally, consideration must be given to the effect (including changes to bioavailability) of fermentation on environmentally acquired (in food and soil) chemical contaminants, such as heavy metals, trace elements, and pesticides. This affects different agro-industries, including the wine industry, in which case pesticides can affect

fermentation due not only to pesticide residues' presence in terms of legal compliance but to its effect on the yeast responsible for the alcoholic fermentation process.

Different studies have shown the antagonistic effect of pesticides over yeasts, mainly for organic pesticides such as Acylalanine, Anilinopyrimidine, Benzimidazole, Dicarboxamide, Pyrethroid, Organochlorine, Phthalimide, Pyrimidine, and Triazole [20,28–31]. Čuš and Raspor [22] observed that in the presence of different concentrations of the fungicide Pyrimethanil (1 and 10 mg/L), non-Saccharomyces yeast such as *Hanseniaspora uvarum* significantly decrease their anaerobic growth rates with both concentrations of the fungicide, confirming the negative effect of this fungicide on its growth. Notably, when Pyrimethanil was applied with *S. cerevisiae*, only the higher rate of it was required to reduce the organism's growth. In the same study, in the case of *S. cerevisiae*, they did not observe an influence of the pesticide on the reducing sugars at the end of cultivation, although an increase in the volatile acidity in wine was reported for both yeasts used, *H. uvarum* and *S. cerevisiae*, reducing the final quality of the wine. The same authors reported that the initial concentration of the yeast in the must was correlated with the effect of Pyrimethanil on wine fermentation; in the case of *H. uvarum*, the inhibitory effect of the pesticide was higher when the lower initial concentration of yeast cells was quantified. All the previous information shows an approximation of how variable the influence of one pesticide can be over different yeasts.

Another study on the negative impacts of pesticides on the normal development of alcoholic fermentation demonstrated that in wine must contaminated with the fungicides Metalaxyl-m and Folpet, the growth of Saccharomyces and non-Saccharomyces yeasts could be diminished, even though their effects were not always at the same rate in comparison with other yeasts [32]. The authors used a commercial formulation containing both the previously mentioned fungicides, named Ridomil Gold® (Syngenta, Milan, Italy), and a mixture of analytical grade standards. The results were variable depending on the yeast used; for instance, in the case of *Torulaspora delbrueckii* B05B12 yeast, Folpet, the mix with analytical standards and the commercial formulation negatively affected the weight reduction in time, but the Metalaxil-m analytical standard had no negative effect, showing similar weight reduction compared to the control. The previous results were different from the ones for *S. cerevisiae* E4, which only showed a negative effect on the weight reduction in time when the commercial formulation was applied; the latter could be showing an effect of the coformulations of the fungicide on the yeast. Bizaj et al. [33] have shown the interactions between different strains of industrial yeasts *S. cerevisiae* and fungicides Fenhexamid and Pyrimethanil present in grape juice and their effect on wine composition profile, demonstrating that these pesticides have a negative effect on the fermentation kinetics, increasing the concentration of undesirable compounds. For example, Pyrimethanil increases the concentration of Ethyl 2-methylpropanoate, which could be an indicator of lower quality in wines. In the case of higher alcohols such as 3-methyl butanol, which confers fruity flavor to wine, it was negatively affected by Pyrimethanil and Fenhexamid, reducing its concentration. Further reports have also displayed difficulties associated with pesticide residues of Mancozeb (F), Captan (F), Lambda-cyhalothrin (I), and Trifloxystrobin (F), causing a decrease in the growth and metabolism of yeasts [34–36]. Gava et al. [27] reviewed the impact of fungicide residues on the alcoholic fermentation of wine in the literature. Firstly, they reported that the most frequently reported fungicides were Azoxystrobin, Boscalid, Cyprodinil, Fenhexamid, Metalaxyl, and Tebuconazole, among others; these are commonly used fungicides in wine grape production, so this result was not rare. Many of the mentioned pesticides were reported in concentrations between 0.1 and 1 mg/kg, and some others such as Folpet and Pyrimethanil in concentrations above 4 mg/kg. The negative effects of the fungicide residues reported by Gava et al. [27] were a reduction of the maximum specific growth rate (Ametoctradin), membrane integrity modification (Captan), modification of the ethanol and glycerol concentration (Captan, Quinoxifen, Tetraconazole, and Mepanipyrim), and slower fermentation rates (Fenhexamid and Metrafenone).

On the contrary, it has also been reported that some pesticides do not have any adverse effect on the fermentation process. For instance, it has been shown that the presence of the insecticides Chlorpyriphos-methyl, Dimethoate, Fenitrothion, Fenthion, Malathion, Methidathion, Parathion, and Quinalphos during alcoholic fermentation does not exhibit any negative influence on the fermentative activity evaluated as yeast number (cell/mL) and $CO_2$ production (indirect weighing) of *S. cerevisiae* 1161 and 234A [37]. The study involved the usage of high amounts of pesticides, which are unusual for real operating conditions, and therefore it has been concluded that it will be difficult to observe pesticide effects on the yeast activities in cellar conditions. Another study with different doses of the fungicides Cyprodinil, Fludioxonil, and Pyrimethanil residues on *Vitis vinifera* white wine inoculated with three *S. cerevisiae* strains suggested that there are significant differences between the acidic fractions of the aroma, as in some alcohols and esters. Still, these differences do not exceed the perception threshold and do not affect the sensorial quality of most of the wine compared with the untreated samples [14].

A similar investigation has shown that the fungicides Pyrimethanil and Fenhexamid are preferentially removed by *S. cerevisiae* strains (ZIM 1927 and EC-1118) in stationary and fermentative stages during the alcoholic fermentation with concentrations ranging between 0.1 and 1 mg/kg, without causing any problems to reach the dryness of the wine, but there were differences in the duration of alcoholic fermentation of 9 days longer in the case of *S. cerevisiae* EC-1118 [24]. Bizaj et al. [33] supported this and showed other findings with the fungicide Fenhexamid that have shown no effect on alcohol production for strains AWRI 0838 and AWRI 1810; nevertheless, there was a negative effect on some metabolic pathways, such as aromatic composition. The fungicide Fenhexamid was observed to be adsorbed on the *S. cerevisiae* cell wall without degradation during the fermentation process [20]. Several other fungicides showed not to affect yeasts' activity, producing no physical, chemical, or organoleptic alterations, but the effect was yeast-species-dependent [31].

Recently, Terpou et al. [38] evaluated the effect of the fungicide Myclobutanil (a chiral triazole fungicide) on the capacity of two strains of *S. cerevisiae* (LMBF-Y 16 and LMBF-Y-18) to carry out alcoholic fermentation and detoxify the medium. Both strains demonstrated high fermentation efficiency; however, the addition of Myclobutanil marginally slowed the conversion of sugar to ethanol. In addition, a non-negligible elimination of Myclobutanil was observed during fermentation, which ranged overall between 5% and 27%, because of the adsorption or degradation capacity of the yeast. The removal of the fungicide was higher in the case of *S. cerevisiae* LMBF-Y 16, and it was observed that the percentage of removal depended on the concentration of Myclobutanil, being higher (23–27%) with lower concentrations (0.1 mg/kg).

All these investigations revealed variable information regarding the effect of pesticide residues on yeast activity, which poses difficulties in drawing an absolute conclusion. Nevertheless, it is possible to conclude that most of the problematic pesticides are old molecules, such as the fungicides Folpet and Captan [29], which are disappearing from the market, even though some are still in use. Another fact is that there are several studies that used high concentrations of pesticides to elucidate the effect that these ones cause on yeast activity, but they could overestimate the problem since pesticide concentrations are not commonly found in wine cellars. New approaches to manage these problems are being taken for newly released molecules, such as considering studies of fermentation interferences, or even the effect on the quality of wine in a pre-market step.

*Molecular Studies*

Studies have also shown that pesticides have posed profound toxicological effects at the gene level on both the wine's Saccharomyces and non-Saccharomyces yeasts during fermentation. For instance, a microarray study has shown that when *S. cerevisiae* is exposed to the herbicide 2–4-d-Dichlorophenoxyacetic acid (2,4-D) and the fungicide Thiram, a complex gene regulation occurs where a set of genes responsible for carbohydrate metabolism, cell stress, and energy generation are upregulated [39–41]. These genes are specifically

regulated by Msn2p and Msn4p transcription factors, which are translocated to the nucleus from cytosol on receiving such toxic stresses, activating the transcription of target genes [40,41]. In another study, it was shown that the herbicide Glufosinate ammonium (GA) reversibly inhibits glutamine synthetase (Gln1) in *S. cerevisiae*, which alters the yeast performance in wine fermentation of grape juice. The biochemical analysis reveals that GA partially inhibits the nutrient-sensing TORC1 pathway by interacting with the TOR1 gene, which affects the growth and longevity of the yeast [42].

Similarly, the 3-Amino-1,2,4-triazole (3-AT) is the component of the herbicide Amitrol that was shown to inhibit histidine biosynthesis in most of the grape fungal microbiota [43], as well as Sulfometuron methyl (SM), which was able to inhibit acetolactate synthase, an enzyme essential for the biosynthesis of leucine, isoleucine, and valine [44,45]. These indicated inhibitors block the amino acid biosynthesis as the empty tRNAs are sensed by Gcn2 kinase, which blocks the general translation initiation. However, Gcn2 kinase promotes the translation of transcription factor Gcn4, which activates the biosynthesis of all proteinogenic amino acids, in a way blocking the general amino acid control system (GAAC), which is the central system for amino acid synthesis [42,44–46]. Global analyses have shown that Gcn2 kinase mutants become hypersensitive towards these inhibitors that also aid in increasing the amino acid biosynthetic genes in a Gcn4 dependent manner, affecting the growth physiology of the yeast [42]. Other inhibitors such as Methionine sulfoximine (MSX) block glutamine synthase (GS), averting the nitrogen assimilation in yeast by inhibiting the glutamine synthesis and freeing ammonium from glutamic acid [42,47]. The glutamine starvation triggers the inhibition of the target of the rapamycin (TOR) kinase pathway, which controls the nitrogen catabolite repression that promotes the accumulation of poor nitrogen sources such as ammonia or devoid of glutamine [48]. Notably, these dietary restrictions often lead the yeast cells to remain in the stationary phase for a longer period, which also affects the yeast physiology, even under winemaking conditions [49]. Therefore, these pesticides and herbicides affect different biochemical processes during winemaking and provide insights into yeast physiology and the relevant metabolic steps.

Sieiro-Sampedro et al. [50] studied the effect of two commercial fungicides containing Mepanipyrim and Tetraconazole on the biosynthesis of aromatic compounds through proteomic and transcriptomic analysis, and the growth of a commercial *Saccharomyces* yeast (Lalvin T73™). Results showed that Tetraconazole had an effect on mitosis and cell cycle progression, as well as downregulation of essential activators of Cdc28p, a cell cycle regulator. In addition, it was discovered that Tetraconazole promotes the overexpression of some genes such as *BAT1*, *PDC1*, *ADH1*, and proteins involved in the biosynthesis of aromatic compounds. Additionally, the presence of this fungicide affected the fermentative activity of *S. cerevisiae* in Garnacha must [51]. A decrease was observed in glucose consumption. Likewise, significant differences were found in some compounds such as methione (24% decrease), fatty acids (23% to 66% increase), and ethyl esters (23% to 145% increase) in musts enriched with the commercial Tetraconazole formulation. Variations in the content of volatile compounds were related to changes in the activity of enzymes involved in the secondary metabolism of yeasts.

Gil et al. [52] studied the gene expression associated with the response mechanism of *S. cerevisiae* BY4741 to the exposure to different concentrations of the fungicide Pyrimethanil (10, 20, and 110 mg/L) during yeast growth in MMB medium. Transcriptomic analyses showed that, overall, 1677 genes were affected, showing that 777 were upregulated and 900 downregulated. The changes in expression were concentration-dependent, and mainly associated with the metabolism of amino acids related to arginine biosynthesis (*ARG1* to *ARG11*, and *CPA1/CPA2*) and the metabolism of sulfur amino acids such as methionine and cysteine.

### 3. Effect of Yeast over Pesticide Dissipation

It is a well-known fact that pesticide residue concentration is considerably diminished during the winemaking process compared with those present in the grapes due to the contribution of different stages within this process, such as pressing and destemming, alcoholic fermentation, clarification, malolactic fermentation (red wine), and filtration [17,18,29,53]. Between the different steps involved in the pesticide dissipation, alcoholic fermentation is probably the main one, where the *Saccharomyces* yeasts and others (non-*Saccharomyces*) transform sugars mainly into alcohol and carbon dioxide, among other compounds formed as by-products. It has been observed that following alcoholic fermentation, pesticide residues have reduced concentrations in wine at variable amounts depending on the yeast and active ingredient. Numerous studies have suggested that the underlying mechanisms could be due to chemical or biological degradation or adsorption with the components of yeast cell walls such as chitin and glucan (Figure 1) [54].

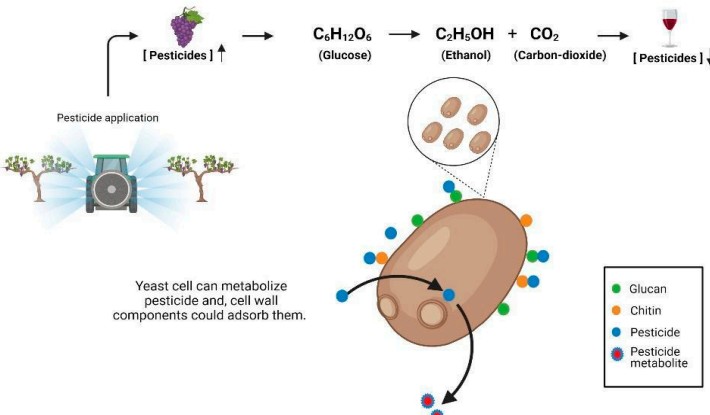

**Figure 1.** Schematic representations of alcoholic fermentation and mechanisms involved in the adsorption of some pesticides in wine.

The previous phenomenon is variable, and does not affect all pesticides equally [55,56]. For instance, some fungicides such as Pyrimethanil, Cyprodinil, and Procymidone have shown different adsorption on lees ($0.90 \pm 0.03$; $0.43 \pm 0.03$; $1.09 \pm 0.05$ mg/L, respectively) [18]. Čuš et al. [17] agreed with another study made by Fernández et al. [57], in which lees adsorption was quantified, revealing the same adsorption phenomenon for Pyrimethanil (F) and Cyprodinil (F), and two other fungicides (Fludioxinil and Quinoxyfen). All the previous articles show different amounts of pesticide removal after alcoholic fermentation, which could be variable, as can be seen in Table 2, ranging from 20 to almost 100 percent for the studied pesticides in these articles.

**Table 2.** Pesticide residues in mg/kg.

| Pesticide | Pre A.F [a] | Wine | Reduction [b] (%) | Refs. |
|---|---|---|---|---|
| **Acetamiprid** | **0.681** | **0.474** | **30.4** | |
| Buprofezin | 4.643 | 1.218 | 73.8 | Alister et al., 2014 |
| Imidacloprid | 3.565 | 2.826 | 20.7 | |
| Lamda-cyhalotrin | 0.02 | 0.003 | 85.0 | |
| Pyrimethanil | 5.084 | 1.313 | 74.2 | |
| Tebuconazole | 7.601 | 0.732 | 90.4 | |
| Boscalid | $4.26 \pm 1.28$ | $1.00 \pm 0.32$ | 76.5 | Angioni et al., 2011 |
| Indoxacarb | $0.54 \pm 0.14$ | <0.018 | 96.7 | |
| Iprovalicarb | $0.79 \pm 0.27$ | $0.36 \pm 0.12$ | 54.4 | |
| Azoxystrobin | 0.13 | 0.09 | 30.8 | Cabras and Angioni, 2000 |
| Cyprodinil | 0.36 | 0.21 | 41.7 | |
| Fludioxonil | 0.39 | <0.05 | 87.2 | |
| Mepanipyrim | 0.16 | <0.01 | 93.8 | |
| Boscalid | $0.02 \pm 0.01$ | $0.01 \pm 0.00$ | 50.0 | Cus et al., 2010 |
| Chlorotalonil | $0.04 \pm 0.01$ | <0.01 | 75.0 | |
| Dimethomorph | $0.05 \pm 0.01$ | $0.008 \pm 0.00$ | 84.0 | |
| Folpet | $0.04 \pm 0.02$ | <0.02 | 50.0 | |
| Procymidone | $0.14 \pm 0.02$ | 0.07 | 50.0 | |
| Pyrimethanil | $0.18 \pm 0.03$ | <0.01 | 94.4 | |
| Chlorpyrifos | $2.005 \pm 0.262$ | $0.027 \pm 0.005$ | 98.7 | Navarro et al., 1999 |
| Fenarimol | $0.828 \pm 0.187$ | $0.143 \pm 0.035$ | 82.7 | |
| Metalaxyl | $0.650 \pm 0.181$ | $0.450 \pm 0.044$ | 30.8 | |
| Penconazole | $0.575 \pm 0.061$ | $0.093 \pm 0.010$ | 83.8 | |
| Vinclozolin | $1.133 \pm 0.042$ | $0.224 \pm 0.020$ | 80.2 | |

[a] Previous alcoholic fermentation; [b] percentage of pesticide reduction after alcoholic fermentation.

The removal of undesirable compounds from must could be influenced by various factors such as the autolysis state of yeast lees, pH, temperature, and others such as ethanol concentration, which can directly affect yeast behavior [58,59]. This influence could be related to a decrease in yeast activity during fermentation, such as the effect of high temperatures that affect the metabolism, or the increase in ethanol concentration affecting the cellular integrity of yeasts. Reduction in pesticide concentration depends on the mechanism by which yeasts would interact with them, being this by adsorption or metabolization. The metabolization of pesticides by yeast could produce as a consequence metabolites that could be more toxic than their parental molecules. Caridi et al. [60] reviewed in the literature the capacity of adsorption or the affinity of different yeast cell wall constituents with some components of wine, such as phenolic compounds and pigments. This phenomenon has also been reported in wine contaminated with ochratoxin A (oA), and the result suggested that oA is adsorbed by mannoproteins through a spontaneous mechanism of adsorption. The ability to adsorb different compounds by yeasts has been shown in several research articles related to different processes. Examples of this are the removal of unwanted compounds from wine, such as volatile phenols and mycotoxins, or even organic amendments to sequester contaminants in soil [58,59,61,62]; however, nothing has been accomplished so far in order to reduce pesticide residues in wine.

While dissipation by yeast strains may vary between pesticides, there are variances among different strains as well [38]. Bizaj et al. [21] reported different removal percentages of the fungicide Fenhexamid by different strains of *S. cerevisiae* ZIM 1927 and EC-1118 (42.2 and 29.2%, respectively). These variances could be due to differences in the cell wall structures, such as mannan, glucan, and chitin content of the different yeast strains, that affect the adsorption ratio. Caboni and Cabras [26] studied the dissipation of several pesticides during alcoholic fermentation, reporting that pesticides such as Carbaril (I) and Viclozonil (F) reduce their concentration mainly by the phenomenon of adsorption by

yeast constituents. The adsorption differences between wine lees have been observed in a study where vineyard soil amended with solid wine lees (centrifuged lees containing >40% organic matter) exhibited an increase in Fenhexamid adsorption compared with the soil amended with whole wine lees (<10% organic matter). Additionally, when different inactivated yeasts *(S. cerevisiae, M. pulcherrima,* and *K. piculate*) were tested for Fenhexamid adsorption, the highest adsorption was obtained with *S. cerevisiae,* followed by *M. pulcherrima* and *K. apiculata* [63].

Cabras et al. [54] evaluated cell wall constituents' capacity to adsorb the fungicide Fenhexamid. The cell wall is mainly composed of chitin and glucan, so the authors used these two separately (1,3-β-D-glucan and Chitin) to test their potential adsorbing capacity on the pesticide mentioned. The results showed that both had a big adsorption capacity due to their functional groups being capable of a xenobiotic binding; further, glucan showed a greater retention capacity compared with chitin. The latter was explained because glucan has more binding sites available, primarily their hydroxyl moieties, that favor hydrogen bonding interactions.

As can be seen in Table 2, among the pesticides reviewed in the different articles concerning dissipation during alcoholic fermentation, some of them dissipate more than 90 percent. Among them, lipophilicity is a similar characteristic, since all of them are lipophilic molecules with a LogKow value higher than 2.5. The previous has been observed by Alister et al. [18], showing that lipophilicity was a major factor in pesticide dissipation in the winemaking process. Doulia et al. [9] reported that the pesticides that were highly removed from wine were those with higher lipophilicity.

There have been attempts to improve the yeasts' capacity to adsorb different wine components; one of them is a study that used a recombinant *S. cerevisiae* strain overproducing mannoproteins, a consequence of the deletion of the gene *KNR4*, to stabilize wine against protein haze [64]. Studies like this last one could also provide an interesting perspective of what can be done to improve the overproduction of other cell wall components in order to reduce pesticides during the fermentation.

## 4. Conclusions and Future Approach

Pesticides are an essential tool to modern agriculture since they provide, in the end, an economic benefit to producers and ensure the supply to consumers. However, their use is highly disputed among consumers, mainly due to the residues present in food or the pollution of the environment. Contrarily, they generally appear in very low concentrations, and cases of exceeding the maximum residue limits are very few in comparison with their use rate. Nowadays, big market suppliers and consumer organizations exist, demanding the reduction of the amount of these residues, with the wine industry being one of the targets of these regulations. The latter is probably because wine consumption is associated with health benefits, the relation with the Mediterranean diet, and the cardiovascular effect of antioxidants in wine, so the presence of pesticide residues would be contrary to the above mentioned.

Among the principal countries affected by these regulations are mostly countries in the global South, such as Chile, which, due to its variable climatic conditions that benefit the development of fungal diseases and pests, implements many pesticides used to produce high-quality food and other agricultural products. For this reason, advancement in the investigations related to the pesticide dissipation process and how to use this information as a direct method to reduce or remove them, particularly in the wine industry, is an important task that has been little investigated and is still pending. The present review article emphasizes only the phenomenon but not its application, such as the capacity of yeast to reduce pesticide residue concentrations, and its use as a strategic tool for the wine industry. It is important to consider the use of yeasts utilized in the wine industry for bioremediation or bio-sorption processes in order to reduce or remove pesticides from the wine matrix. Modern agricultural practices should search for and implement new techniques which can eventually allow the wine industry, and even other agro-industries,

to develop strategies that help to reduce or remove the pesticide residues in/on the food, as a way to improve their processes and provide safe and healthy food products to consumers.

Nowadays, the most important gap in reducing pesticide content in wine is the low tolerance by consumers, who demand zero residues in wine. The different techniques studied to remove pesticides from wine show some problems with effectiveness and specificity. For instance, bentonite and activated carbon could diminish the aroma and color of wine. Additionally, using energy for tools such as pulsed electric field and ultrasound represents a higher cost of production. Finally, the scaling of new technologies could also be a problem.

On the other hand, defining a yeast strain as the best at lowering the pesticide content is not possible because the conditions in which the different investigations have been developed are very dissimilar regarding the media, pesticide concentration, temperature, duration, and yeast strain. Investigation regarding this premise could result in very interesting and high-impact information, and it is without a doubt one of the goals of our future investigations.

More information and new studies are needed, such as a selection of yeasts by their dissipation capacity or pesticide removal. Technics such as adaptive laboratory evolution (ALE) could be used to improve the yeast removal capacity. There are several examples of the use of ALE to improve different yeast capacities such as ethanol production increase [65], glycerol production enhancement [66], carotenoid production [67], high-temperature tolerance [68], and greater production of fermentative aroma [69]. Studies that allow characterizing the particular yeast affinity to remove pesticides should be also developed. This will permit the selection of yeast from a vast pool of yeast strains currently available.

**Author Contributions:** Writing—original draft preparation, K.B. and L.G.; writing—review and editing, K.B., S.G. and L.G.; visualization and funding acquisition, L.G. All authors have read and agreed to the published version of the manuscript.

**Funding:** This research was funded by ANID/CONICYT FONDECYT Iniciación 11180979 and ANID Doctoral Fellowship Support Grant 24210342.

**Institutional Review Board Statement:** Not applicable.

**Informed Consent Statement:** Not applicable.

**Acknowledgments:** Not applicable.

**Conflicts of Interest:** The authors declare no conflict of interest.

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
