# Peer review of "Pesticide and Yeast Interaction in Alcoholic Fermentation: A Mini-Review"

_fermentation, doi:10.3390/fermentation9030266_

Round 1

Reviewer 1 Report

The manuscript entitled “Pesticide and yeast interaction in alcoholic fermentation: A mini-review”  is consistent with the scope of the Journal. Unfortunately in the current state, there are several weaknesses points in the manuscript that the authors should further reconsider and clarify.  What is the novelty of this study? Can new solutions be introduced ? The manuscript needs to be supported by more recent scientific publications. Only 6 references out of 55 are from the year above 2015 years. A high-quality papers has to provide a proper state-of-the-art analysis. Literature data should have been compared to the newest research. Be consistent regarding units. Adapt them to the requirements of the journal. A through revision is required prior to publication.

Author Response

The manuscript entitled “Pesticide and yeast interaction in alcoholic fermentation: A mini-review”  is consistent with the scope of the Journal. Unfortunately in the current state, there are several weaknesses points in the manuscript that the authors should further reconsider and clarify.  

Q1. What is the novelty of this study?

R1. Thank you for the positive comments. The manuscript has been changed to include observations on the Abstract.

Q2.Can new solutions be introduced ? 

 R2. We included information related to that point. Lines 425-433

Q3.The manuscript needs to be supported by more recent scientific publications. Only 6 references out of 55 are from the year above 2015 years. A high-quality papers has to provide a proper state-of-the-art analysis. Literature data should have been compared to the newest research. Be consistent regarding units. Adapt them to the requirements of the journal. A through revision is required prior to publication.

R3. We improve and included new references in the manuscript. Lines 77-90. Also, we changed the units based on SI.

Reviewer 2 Report

The article presents the pesticide effect on yeast during wine - alcoholic fermentation. It is an interesting article, but it requires several modifications:

The bibliographic information is quite old. Please check some newer information:  https://doi.org/10.1016/j.foodcont.2021.108534; https://doi.org/10.1016/j.jfca.2022.104714; https://doi.org/10.1016/j.foodcont.2016.07.035, https://doi.org/10.1016/j.envpol.2021.116827, 10.5772/intechopen.95210 (Chapter 2)

The Abstract must be rewritten, as it is difficult to follow: ", then it is reviewed some examples of...", " the study reveals the underlying mechanisms involved..."

Introduction: Should be modified: it needs to present the structure of the review; a table presenting some example of pesticides presents in wine and their impact on wine should be added, also some regulations in terms of maximum acceptance should be provided. 

Chapter 2. Several English mistakes need to be solved: please rephrase: 72 These removal processes would be due to adsorption or degradation being the first of these, probably the main one"; 111 The authors carried out fermentation assays with the commercial formulation of Ridomil Gold® (Syngenta) containing both fungicides previously mentioned...; 113 In addition, studied the individual effect of the active ingredients of Ridomil (Metalaxyl-m and Folpet) and the mix of them but using analytical grade standards.;

The discussion should be improved by presenting the compounds as classes with the same structure or at least property - insecticides, rodenticides, herbicides, fungicides, biocides. Present shortly the main yeast strains used for wine making and the differences between them. 

Chapter 3: please rephrase: 276The reasons for the reduction in  concentration will be depending on the mechanism that yeast could affect pesticide, being by adsorption or by metabolization with the consequence of metabolite production.

What are the various techniques for pesticide degradation? A comprehensive list of tables should be provided either based on pesticide type or on yeast type.

What parameters affect the pesticides reduction process?

What are gaps in reducing the pesticide content in wine?

Which yeast strain is the best at lowering the pesticide content?

What are the authors recommendations? 

Future recommendations are a critical part of the review. It should be provided.

Author Response

The article presents the pesticide effect on yeast during wine - alcoholic fermentation. It is an interesting article, but it requires several modifications.

Thank you for the positive comments.

Q1. The bibliographic information is quite old. Please check some newer information:  https://doi.org/10.1016/j.foodcont.2021.108534; https://doi.org/10.1016/j.jfca.2022.104714; https://doi.org/10.1016/j.foodcont.2016.07.035, https://doi.org/10.1016/j.envpol.2021.116827, 10.5772/intechopen.95210 (Chapter 2).

R1. We improve and  included new reference. Lines  77-90.

Q2. The Abstract must be rewritten, as it is difficult to follow: ", then it is reviewed some examples of...", " the study reveals the underlying mechanisms involved..."

R2. The Abstract was modified following your comments.

Q3.Introduction: Should be modified: it needs to present the structure of the review; a table presenting some example of pesticides presents in wine and their impact on wine should be added, also some regulations in terms of maximum acceptance should be provided. 

R3. Introduction was improved and we included information related to maximum Residue Limit in wine grapes for different pesticides. Lines 49-68

Q4.Chapter 2. Several English mistakes need to be solved: please rephrase: 72 These removal processes would be due to adsorption or degradation being the first of these, probably the main one"

R4. This was changed. Lines 115-116

Q5. 111 The authors carried out fermentation assays with the commercial formulation of Ridomil Gold® (Syngenta) containing both fungicides previously mentioned...

R5. This was corrected. Lines 155-157.

Q6. 113 In addition, studied the individual effect of the active ingredients of Ridomil (Metalaxyl-m and Folpet) and the mix of them but using analytical grade standards.

R6. This was improved. Lines 155-157

Q5.The discussion should be improved by presenting the compounds as classes with the same structure or at least property - insecticides, rodenticides, herbicides, fungicides, biocides. Present shortly the main yeast strains used for wine making and the differences between them. 

R5. We individualize through all manuscripts the property of compounds. Also, we added characteristics of the main wine yeast used. Lines 103-106.

Q6. Chapter 3: please rephrase: 276The reasons for the reduction in  concentration will be depending on the mechanism that yeast could affect pesticide, being by adsorption or by metabolization with the consequence of metabolite production.

R6. We rephrased it. Lines 334-337.

Q7.What are the various techniques for pesticide degradation? A comprehensive list of tables should be provided either based on pesticide type or on yeast type.

R7. We mentioned in the manuscript the main process associated to pesticide degradation.

Q8.What parameters affect the pesticides reduction process?

R8. This was included. Lines 329-334.

Q9.What are gaps in reducing the pesticide content in wine?

R9. This was included. Lines 412-418.

Q10.Which yeast strain is the best at lowering the pesticide content?

R10. This was included. Lines 419-424.

Q11.What are the authors recommendations? 

R11. This was included. Lines 425-433

Q12.Future recommendations are a critical part of the review. It should be provided.

R12. This was included. Lines 412-433.